# A Novel POP-Ni Catalyst Derived from PBTP for Ambient Fixation of CO_2_ into Cyclic Carbonates

**DOI:** 10.3390/ma16062132

**Published:** 2023-03-07

**Authors:** Fen Wei, Jiaxiang Qiu, Yanbin Zeng, Zhimeng Liu, Xiaoxia Wang, Guanqun Xie

**Affiliations:** 1Guangdong Provincial Engineering Technology Research Center of Key Material for High Performance Copper Clad Laminate, School of Materials Science and Engineering, Dongguan University of Technology, Dongguan 523808, China; 2School of Environment and Civil Engineering, Dongguan University of Technology, Dongguan 523808, China

**Keywords:** heterogeneous catalysis, POP-Ni catalyst, carbon dioxide fixation, green chemistry, catalytic material

## Abstract

The immobilization of homogeneous catalysts has always been a hot issue in the field of catalysis. In this paper, in an attempt to immobilize the homogeneous [Ni(Me_6_Tren)X]X (X = I, Br, Cl)-type catalyst with porous organic polymer (POP), the heterogeneous catalyst PBTP-Me_6_Tren(Ni) (POP-Ni) was designed and constructed by quaternization of the porous bromomethyl benzene polymer (PBTP) with tri[2-(dimethylamino)ethyl]amine (Me_6_Tren) followed by coordination of the Ni(II) Lewis acidic center. Evaluation of the performance of the POP-Ni catalyst found it was able to catalyze the CO_2_ cycloaddition with epichlorohydrin in N,N-dimethylformamide (DMF), affording 97.5% yield with 99% selectivity of chloropropylene carbonate under ambient conditions (80 °C, CO_2_ balloon). The excellent catalytic performance of POP-Ni could be attributed to its porous properties, the intramolecular synergy between Lewis acid Ni(II) and nucleophilic Br anion, and the efficient adsorption of CO_2_ by the multiamines Me_6_Tren. In addition, POP-Ni can be conveniently recovered through simple centrifugation, and up to 91.8% yield can be obtained on the sixth run. This research provided a facile approach to multifunctional POP-supported Ni(II) catalysts and may find promising application for sustainable and green synthesis of cyclic carbonates.

## 1. Introduction

In recent years, the issue of persistent increases in CO_2_ in the atmosphere caused by the massive combustion of fossil fuels has become one of the most concerns around the world, and solutions to the issue have been paid significant attention to by both governments and researchers all over the world [1,2,3]. A number of measures have been taken to reduce the release of CO_2_. Alternatively, utilization of CO_2_ as an abundant, non-toxic, and inexpensive C1 source could provide a practical solution. As has been reported, CO_2_ can be converted into alcohols [4,5], acids [6], esters [7,8], polycarbonates [9], and other high-value chemicals [10,11]. It is worth noting that among various chemical fixation methods of CO_2_, the cycloaddition of CO_2_ and epoxide to produce cyclic carbonate has attracted widespread attention from many researchers [12,13,14]. On the one hand, the cycloaddition reaction of CO_2_ and epoxide is more beneficial to the development of green chemistry due to its 100% atomic economy [15]; on the other hand, as an important type of chemical product, cyclic carbonates have been commonly applied as fuel additives [16], polar non-protonic solvents [17], battery electrolytes [18], useful intermediates of drugs/fine chemicals [19], and the monomers for polycarbonate and polyurethane [20,21]. However, owing to the high thermodynamic and kinetic stability of CO_2_, its conversion to cyclic carbonate is highly energy-consuming and requires the use of an active catalyst. Hitherto, there have been many catalytic systems available for the transformation of CO_2_ [22,23]. Among them, metal complexes have stood out in many catalytic systems due to their high activities, simple preparative procedures, as well as low-cost starting materials [24,25]. The metal complex catalytic system may be divided into binary catalytic systems and one-component bifunctional catalysts. As for the former, the metal complex catalysts have to be assisted by additional co-catalysts such as tetrabutylammonium bromide to afford satisfactory performance [14,26]. Recently, more and more efforts have been focused on the development of one-component bifunctional metal catalyst systems that integrate metal Lewis sites and nucleophilic sites in one component [27,28]. The development of one-component bifunctional catalysts could not only eliminate the extra addition of co-catalysts, but also increase the intramolecular cooperation between the metal Lewis acidic sites and nucleophilic sites in the catalysts, thereby enhancing the catalytic activity. In this context, Naveen [29] and our group [15] reported a series of one-component bifunctional catalysts [M(Me_6_Tren)X]X. Among them, [Ni(Me_6_Tren)X]X (X = I, Br, Cl) exhibited the best catalytic activity for the efficient conversion of CO_2_ and epoxide to cyclic carbonate under mild conditions. However, recovery of these homogeneous [Ni(Me_6_Tren)X]X catalysts was challenging due to its cumbersome recovery procedures having a large amount of organic solvent and incomplete recovery. In view of the recoverability of heterogeneous catalysts [30,31,32,33], which could maintain the good catalytic activity of the counterpart homogeneous catalysts and have better potential for practical application, we were inspired to attempt the heterogenization of the [Ni(Me_6_Tren)X]X catalysts as a novel heterogeneous catalyst.

PBTP-(x)-R, a series of novel amine-containing porous organic copolymers synthesized by Yang and co-workers [34] via copolymerization of 4,4’-bis (chloromethyl) biphenyl (BCMBP) and 1,3,5-tris (chloromethyl) benzene (TCB) followed by reactions with multiamines with benzyl chloride functionality, showed selective-adsorption and high-adsorption capacity for CO_2_. In combination with our previous work where [Ni(Me_6_Tren)X]X (X = I, Br, Cl) could be an efficient homogeneous catalyst for the fixation of CO_2_ into cyclic carbonates, herein a novel heterogeneous PBTP-Me_6_Tren(Ni) (POP-Ni) catalyst was designed and constructed by grafting tri[2-(dimethylamino)ethyl]amine (Me_6_Tren) on a porous organic polymer PBTP followed by coordination with Ni(II) salt, envisioning that it would be able to enrich CO_2_ and catalyze CO_2_ cycloaddition reaction efficiently.

## 2. Materials and Methods

### 2.1. Chemicals and Instruments

1,3,5-Tris(bromomethyl) benzene (TBB) (98%), 1,3,5-trimethoxybenzene (98%), tri[2-(dimethylamino)ethyl] amine (Me_6_Tren) (98%), epichlorohydrin (98%), epibromohydrin (97%), styrene oxide (98%), butyl glycidyl ether (98%), *tert*-butyl glycidyl ether (97%), allyl glycidyl ether (98%), phenyl glycidyl ether (97%), and 1,2-epoxyhexane (97%) were obtained from Energy Chemical (Anhui, China). 4,4’-(Bromomethyl) biphenyl (BBMBP) (97%) was purchased from TCI (Tokyo, Japan). Ferric chloride (95%) was purchased from 3A Chemicals (Shanghai, China). Both 1,2-dichloroethane and nickel acetate tetrahydrate (98%) were purchased from Macklin Chemicals (Shanghai, China). All reagents used in this experiment were directly used without any pretreatment.

The surface morphology of the sample was characterized by means of scanning electron microscopy (SEM) on a JSM-6701F scanning electron microscope produced by Japan Electronics Co., Ltd. (Tokyo, Japan). The N_2_ sorption isotherms were performed on Quantachrome-EVO (Quantachrome, Hillsboro, OR, USA), and the sample powder was degassed at 150 °C in a vacuum for 12 h before measurement. By using Thermo Scientific K-Alpha (Thermo Fisher Scientific, Waltham, MA, USA), X-ray photoelectron spectroscopy (XPS) characterization was performed for the sample. Thermogravimetric analysis (TGA) was conducted by using Netzsch TG209-F3 (Netzsch, Selbu, Germany) in an N_2_ atmosphere. The reaction yield was detected using the Shimadzu GC2010-QP2010Plus gas chromatography mass spectrometer (GC-MS) (Shimadzu, Japan) with the gas chromatography column of Agilent J&W HP-5, and the GC column temperature was programmed at 32–150 °C at a rate of 8 °C/min.

### 2.2. Synthesis of PBTP

PBTP was synthesized by referring to Yang’s work [34]. A mixture of 4,4′-(bromomethyl) biphenyl (BBMBP, 0.340 g), and 1,3,5-tris (bromomethyl) benzene (TBB, 1.427 g) was completely dissolved in 1,2-dichloroethane (30 mL). Under an N_2_ atmosphere, anhydrous FeCl_3_ (0.811 g) was added swiftly into the above solution. After being stirred at 45 °C for 1 h, the reaction system was heated to 80 °C for another 1 h. After the completion of the reaction, the solid reaction product was thoroughly washed with methanol and dried in a vacuum at 70 °C for 24 h to obtain dark-brown solid powder.

### 2.3. The Synthesis of PBTP-Me_6_Tren

PBTP (0.1 g) was added into a Pyrex tube (15 mL), and toluene (2.5 mL) was added for its uniform distribution. Then, tri[2-(dimethylamino)ethyl] amine (Me_6_Tren, 0.2 mmol, 0.11 mL) was added dropwise under stirring. After the complete dripping of Me_6_Tren, the screw plug of the Pyrex tube was immediately fitted. Vigorous stirring was performed at 90 °C for 72 h. After the reaction, the reaction product was fully washed with methanol and then dried in a vacuum at 60 °C for 12 h to obtain a dark-brown powder.

### 2.4. The Synthesis of PBTP-Me_6_Tren (Ni) (POP-Ni)

Nickel acetate tetrahydrate (2 mmol, 2.488 g) was dissolved in N, N-dimethylformamide (DMF, 25 mL) and filtered to obtain a clear solution. Then, PBTP-Me_6_Tren (0.1 g) was uniformly distributed in the above solution, and the reaction system was allowed to react under reflux at 90 °C for 6 h. After the reaction, the reaction product was washed thoroughly with a large amount of methanol and dried in a vacuum at 60 °C for 12 h to obtain a dark-brown powder.

### 2.5. Evaluation of the Catalyst by CO_2_ Cycloaddition Reaction

Typically, a Schlenk tube (10 mL) containing the catalyst POP-Ni (80 mg) was emptied and backfilled with CO_2_ three times. Under CO_2_ balloon pressure, DMF (0.5 mL) and epichlorohydrin (5 mmol, 0.394 mL) were added via a syringe. Then, the mixture was stirred at 80 °C for 24 h. The crude reaction mixture was centrifuged, and the up-layer solution was then diluted with ethyl acetate. The yield was determined by GC-MS using 1,3,5-trimethoxybenzene as an internal standard. The low-layer catalyst was repeatedly washed with ethanol several times. It was used for the next run after drying in a vacuum at 70 °C for 6 h. The same conditions were used for the reusability studies.

## 3. Results and Discussion

### 3.1. Catalyst Characterization

Porous organic polymer PBTP was synthesized through the copolymerization of monomer 4,4′-(bromomethyl) biphenyl (BBMBP) and 1,3,5-tris (bromomethyl) (TBB) via a Friedel–Crafts alkylation reaction, and then the bifunctional catalyst PBTP-Me_6_Tren(Ni) (POP-Ni) was synthesized by following a two-step sequential post-synthetic modification procedure. (Figure 1). Firstly, the surface morphology of the synthesized material was characterized by scanning electron microscopy (SEM). As shown in Figure 2a, the synthetic PBTP is formed by the accumulation of a spherical-like structure with a rich pore structure. After post-synthetic modification, the surface morphology of POP-Ni shows no obvious change, and it remains an accumulation of a spheroid-like structure. However, compared with PBTP, the size of the spheroid-like structural unit of POP-Ni shows a significant increase (increase from 106.7 ± 19.9 nm to 153.7 ± 21.5 nm) (Figure 2b and Appendix A), indicating that the post-synthetic modification strategy follows a core–shell growth process based on PBTP. To further explore the impact of post-synthetic modification strategies on the pore size distribution of materials, the specific surface area and pore size distributions of PBTP and POP-Ni were examined by means of Brunauer–Emmett–Teller (BET) measurements (Figure 2c,d). According to the measurement results, PBTP has a large specific surface area of up to 902.3 m^2^g^−1^. In terms of pore size distribution, the pore sizes of PBTP concentrates were in the range of 0.8–2.0 nm and 3.2–27 nm. Following the process of two-step sequential post-synthetic modification, the specific surface area of material POP-Ni was reduced from 902.3 m^2^g^−1^ to 576.3 m^2^g^−1^. Meanwhile, compared with PBTP, the pore size distribution of POP-Ni decreased significantly in the range of 0.8–2.0 nm, which may be the leading cause for the decrease in the specific surface area. On the other hand, there is no significant difference in the pore size distribution of POP-Ni as compared with that of PBTP in the range of 3.2–27 nm. Overall, the presence of micropores in the POP-Ni structure could facilitate CO_2_ enrichment in the material, while the extensive distribution of mesoporous could facilitate product transfer.

Furthermore, XPS was employed to examine the element composition and valence distribution of the bifunctional catalysts. As indicated by the XPS spectrum of POP-Ni, there are four different elements in the material, namely Ni, C, N and Br (Appendix A). As shown in the Ni 2p XPS spectrum of POP-Ni (Figure 3a), the two peaks at around 855.68 eV and 873.33 eV are assigned to Ni^2+^ 2p3/2 and Ni^2+^ 2p1/2, respectively, while the satellite peaks of Ni 2p3/2 at 861.21 eV and the peaks at 878.81 eV are assigned to the satellite peak of Ni 2p1/2. In the Br 3d spectrum (Figure 3b), the peak at 68.19 eV is assigned to the quaternary ammonium bromide anion, and the peak with a binding energy of 70.84 eV results from methyl bromide, indicating an incomplete conversion of the benzyl bromide groups in the porous polymer precursor into the nucleophilic Br anion through a quaternization reaction. The incomplete quaternization reaction may be caused by the steric hindrance and pore size limitation of PBTP material. In the C1s spectrum (Figure 3c), there are three different forms of carbon observed, with the binding energy of 283.85 eV and 286.53 eV related to phenyl carbon and aliphatic chain skeleton carbon, respectively. In addition, the peak at 285.49 eV is assigned to the carbon of −C−N−. In the N 1s spectrum of POP-Ni (Figure 3d), the peaks at 398.64 eV, 400.55 eV, and 406.21 eV are ascribed to −N−C−, N−Ni, and quaternary ammonium N cations, respectively. Then, TGA was applied to evaluate the thermal stability of the synthetic bifunctional POP-Ni catalyst (Appendix A). The weight loss of POP-Ni before 150 °C should result from the volatilization of the DMF solvent, and the catalyst maintains sufficient thermal stability before 250 °C. In summary, the above results demonstrate the successful preparation of the bifunctional catalyst POP-Ni.

### 3.2. Investigation of the Catalytic Performance

The bifunctional catalyst has attracted increased attention due to two advantages: acid-base coordination and easy reusability. In addition, there has been some attention paid in recent years to the research of more moderate reaction conditions and a wide range of substrates. Therefore, the conditions of CO_2_ cycloaddition reactions were examined in this study by using POP-Ni as a catalyst and epichlorohydrin as a template substrate (Table 1). Firstly, the catalytic activity of catalysts PBTP, PBTP-Me_6_Tren, and POP-Ni for cycloaddition reaction was explored under solvent-free conditions with a catalyst loading of 40 mg (Table 1, Entries 1–3). The results show that PBTP as a carrier plays little role in the catalytic activity for CO_2_ cycloaddition reactions. Different from PBTP, the PBTP-Me_6_Tren with nucleophilic halogen active sites and multiamine groups converted epichlorohydrin into chloropropene carbonate with a 32.8% yield. In contrast, the bifunctional catalyst POP-Ni afforded epichlorohydrin with a 51.9% yield, suggesting Ni(II) as the active site of Lewis acid to synergize cycloaddition with nucleophilic Br. These results clearly indicate the important role played by both the intramolecular synergy of metal Ni(II) with Lewis acidity and the nucleophilic halogen Br and the effective adsorption of multiamine groups on CO_2_ [34] in the occurrence of CO_2_ cycloaddition reactions.

To further improve the catalyst activity, three highly polar solvents, DMSO, DMA, and DMF, were introduced into the CO_2_ cycloaddition reaction (Table 1, Entries 4–6) since solvent-free conditions seemed to not disperse the catalyst well. The yield of chloropropene carbonate reached 58.4% when DMSO was used as a solvent, while DMA and DMF as the solvent could afford 69.6% and 78.5% yield, respectively. Although DMSO shows a higher polarity to facilitate the dispersion and swelling of the catalysts, its higher viscosity fails to enhance its catalytic activity significantly compared with DMA and DMF [35]. DMF as a solvent achieves a higher catalytic yield than DMA. The primary reason for this is that the DMF with higher polarity produces a more significant swelling effect on the catalyst, which allows POP-Ni to be better distributed in the reaction system, thus enhancing its catalytic activity.

The influence of the catalyst loading on the product yields was further investigated (Appendix A). According to the results, the catalytic activity was boosted significantly with the increase in catalyst loading: 40 mg (78.5%) < 50 mg (82.3%) < 60 mg (88.2%) < 70 mg (91.2%) < 80 mg (97.5%) < 90 mg (98.7%). The results showed 90 mg of the catalyst did not bring about significant improvement in yields and 80 mg of the catalyst could already afford satisfactory yields (97.5%); thus, 80 mg was set as the optimal catalyst dosage (Table 1, Entry 7). It should be noted that POP-Ni catalyst at 80 °C and CO_2_ balloon conditions showed slightly lower catalytic activity (80 mg catalyst loading, 97.5% yield) compared with that of the homogeneous catalyst [Ni(Me_6_Tren)Br]Br (1 mol% catalyst loading, 98.9% yield), and the selectivity in both cases were excellent (>99%).

In addition, a comparison of POP-Ni with other available POP-related catalysts for the CO_2_ cycloaddition reaction of epichlorohydrin was also made (Table 2). Although mild and efficient catalysts involving Zn or Co Lewis centers on the POP have been developed [36,37,38], the addition of TBAB as a cocatalyst was required, which may make the recovery of catalytic systems tedious and difficult. For other Mg^2+^-, Zn^2+^-, Co^2+^-, or Al^3+^-involved POP catalysts [39,40,41,42,43,44,45], higher temperatures or pressures were usually applied so that comparable yields could be obtained.

### 3.3. Applicability of Catalyst POP-Ni

To illustrate the application scope of POP-Ni, the cycloaddition of CO_2_ with other epoxides was examined under the same conditions (CO_2_ balloon, 80 °C, Table 3, and Appendix A). POP-Ni can also catalyze epibromohydrin with high catalytic activity to generate bromopropylene carbonate with good yields (95.7%) (Table 3, entry 2). For aliphatic epoxides, the yield of cyclic carbonate declines sharply with the increase in the alkyl chain. For 1,2-epoxyhexane, the corresponding cyclic carbonate can be obtained with a medium yield of 62.7% (Table 3, entry 3). With further increases in the length of the alkyl chain, the catalytic activity of POP-Ni was further reduced, as can be seen for the allyl glycidyl ether and butyl glycidyl ether, which afford the yield of 48.0% and 49.2%, respectively (Table 3, entries 4, 5). In addition, the yield of *tert*-butyl glycidyl ethers containing branched alkanes was found to be even lower (35.6%) compared to that of the butyl glycidyl ethers containing straight-chain alkanes (Table 3, entry 6). For phenyl glycidyl ether, it can be transformed into the corresponding cyclic carbonate with a moderate yield of 51.5%. In addition, POP-Ni afforded a yield of 48.2% of the cyclic carbonate when styrene oxide was used as the substrate. It can be seen from the above results that POP-Ni shows good catalytic activity for epichlorohydrin and epichlorohydrin, and its catalytic activity is on a medium level for substrates of relatively larger sizes. This is mainly because the pores in the catalyst are mainly micropores, which restricts the contact between the larger substrate and the catalytic active site of the catalyst and shows certain substrate selectivity.

### 3.4. Catalyst Reusability

Recycling experiments were conducted to investigate the recoverability and stability of the catalyst since they are considered the most important indicators of excellent heterogeneous catalysts. With epichlorohydrin as the substrate, recycling of the catalyst for six runs led to a slight reduction in the catalytic activity of the catalyst (from 97.5% to 91.8%) while maintaining excellent selectivity (>99%) (Figure 4 and Appendix A). At each run of the reaction, the catalyst was easily separated from the reaction system by centrifugation and used for the next run. BET measurement of the recycled catalyst (Appendix A) showed the specific surface area of POP-Ni obviously decreased (403.7 m^2^g^−1^) as compared with the fresh catalyst (576.3 m^2^g^−1^), due mainly to the decrease in micropore numbers, which may probably be caused by blockage of the product molecules in the channel. Overall, the results showed that the mild catalytic condition (80 °C) exerts no obvious effect on the structure and performance of the catalyst, and the catalyst exhibited good potential for practical application.

### 3.5. Plausible Reaction Pathway

Based on the previous reports [44,46] and our experimental results, the reaction pathway of CO_2_ cycloaddition catalyzed by POP-Ni was proposed (Figure 5). Firstly, the epoxide was activated through the Ni^2+^ active site in the catalyst to coordinate with the O atom of the epoxide. Then, the nucleophilic Br anion attacked the carbon atom on the side of the epoxy with less hindrance through a nucleophilic attack to open up the ring (I), thus obtaining the ring-opening O anion intermediate (II). Intermediate II further launches nucleophilic attacks on CO_2_ (III) to obtain carbonate intermediate (IV). Finally, the product cyclic carbonate is obtained through intramolecular substitution. At the same time, the catalyst is released for further catalysis.

## 4. Conclusions

A novel heterogeneous POP-Ni catalyst has been successfully constructed by grafting Me_6_Tren on the PBTP, synthesized by copolymerization of BBMBP and TBB in a molar ratio of 1/4 and subsequent coordination of the Ni(II) Lewis acidic center. The POP-Ni not only possessed a large specific surface area (576.3 m^2^g^−1^) and good thermal stability, but also maintained great catalytic activity after heterogenization. It exhibited excellent catalytic performance with a yield of 97.5% in converting epichlorohydrin and CO_2_ into propylene chlorocarbonate under mild conditions (80 °C, CO_2_ balloon). The excellent catalytic performance of POP-Ni could be attributed to its porous properties, the synergism between the Lewis acid Ni(II) and nucleophilic Br anion, and the efficient adsorption of CO_2_ by the multiamines Me_6_Tren. In particular, POP-Ni could be effectively recovered through simple centrifugal and still maintain excellent catalytic performance with a yield of 91.5% after six consecutive recycles. The POP-Ni catalyst may find promising application in view of its availability, high activity, and good reusability.

## Figures and Tables

**Figure 1 materials-16-02132-f001:**
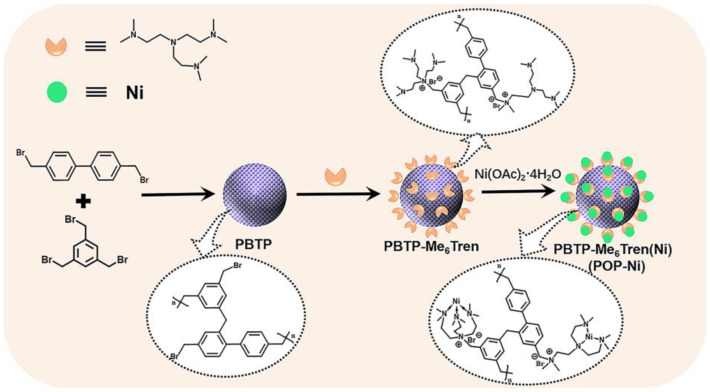
The preparation of PBTP-Me6Tren(Ni) (POP-Ni).

**Figure 2 materials-16-02132-f002:**
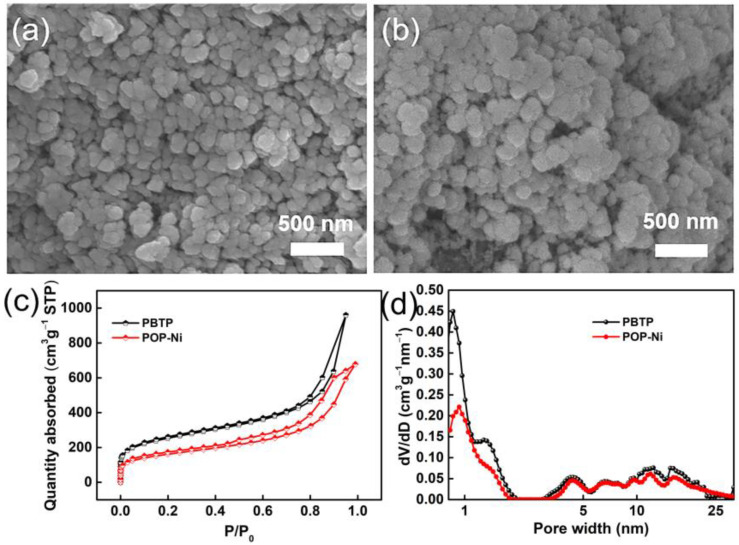
SEM image of POP-Ni (**a**) and POP-Ni (**b**); N_2_ adsorption/desorption isotherm (**c**) and pore size distribution of PBTP and POP-Ni (**d**).

**Figure 3 materials-16-02132-f003:**
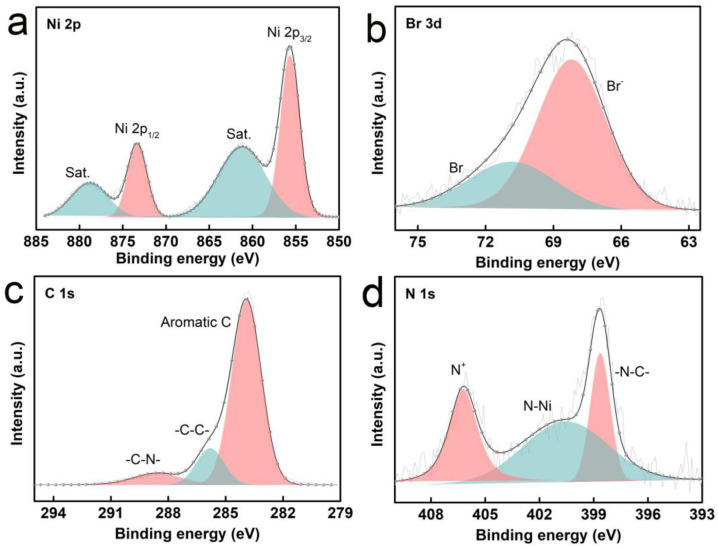
Ni 2p spectrum (**a**); Br 3d spectrum (**b**); C 1s spectrum (**c**); N 1s spectrum of POP-Ni (**d**).

**Figure 4 materials-16-02132-f004:**
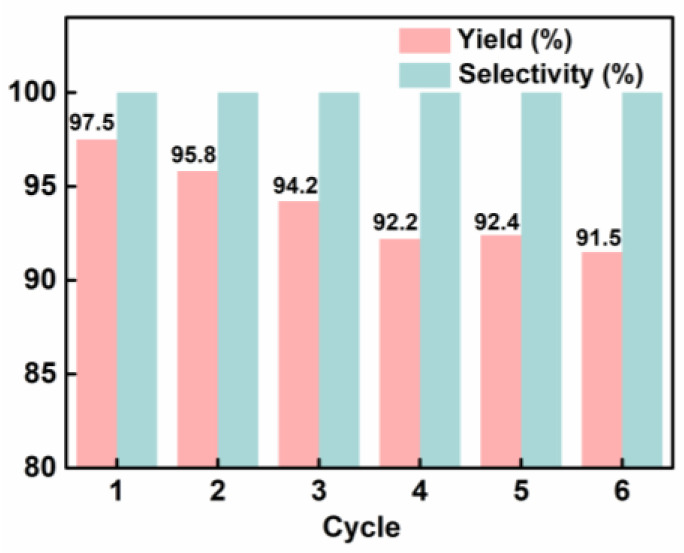
Recycling of the catalyst POP-Ni. Reaction conditions: epichlorohydrin (5 mmol), POP-Ni (80 mg), 0.5 mL of DMF, CO_2_ balloon. The mixture was stirred at 80 °C for 24 h. Yield and selectivity were determined by GC-MS.

**Figure 5 materials-16-02132-f005:**
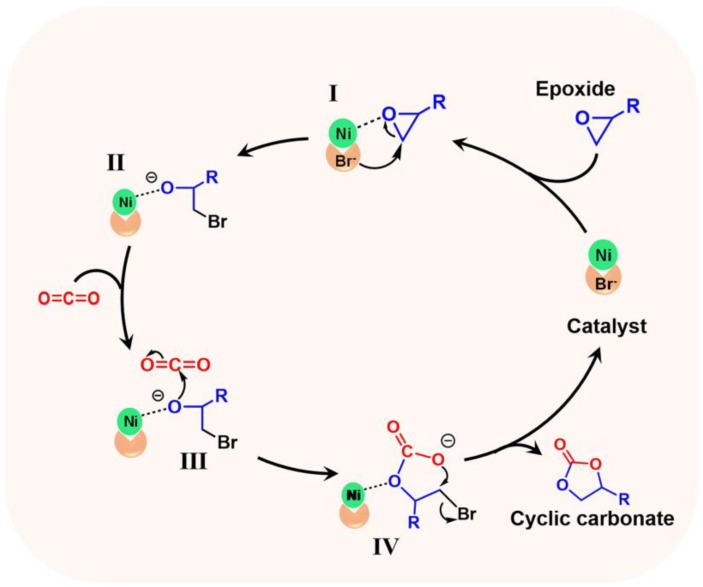
Plausible mechanistic pathway for CO_2_ cycloaddition with epoxides.

**Table 1 materials-16-02132-t001:** Evaluation of the catalytic performance of POP-Ni for CO_2_ cycloaddition reactions ^a^.

	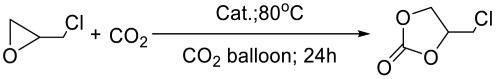
Entry	Catalyst	Amount of Catalyst	Solvent	Yield (%) ^c^	Selectivity (%) ^c^
1	PBTP	40 mg	none	2	>99
2	PBTP-Me_6_Tren	40 mg	none	32.8	>99
3	POP-Ni	40 mg	none	51.9	>99
4	POP-Ni	40 mg	DMSO	58.4	>99
5	POP-Ni	40 mg	DMA	69.6	>99
6	POP-Ni	40 mg	DMF	78.5	>99
7	POP-Ni	80 mg	DMF	97.5	>99
8 ^b^ [15]	[Ni(Me_6_Tren)Br]Br	22 mg	none	98.9	>99

^a^ Reaction conditions: 5 mmol of epichlorohydrin, 0.5 mL of solvent, CO_2_ (balloon), 80 °C, and 24 h. ^b^ The performance of the homogeneous catalyst. Reaction conditions: 5 mmol of epichlorohydrin, 1 mol% of catalyst, CO_2_ (balloon), 80 °C, and 24 h. ^c^ Determined by GC-MS.

**Table 2 materials-16-02132-t002:** The comparison of POP-Ni with available POP related catalysts for the CO_2_ cycloaddition reaction of epichlorohydrin.

	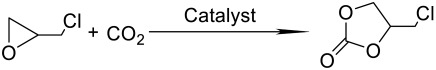
Entry	Catalyst	Cocatalyst	CO_2_ (Mpa)	Temp. (°C)	Time (h)	Yield (%)	Number of Recycling (yield, %) ^a^	Ref.
1	Co/POP-TPP	TBAB	0.1	29	24	95.6	18 (93.6)	[36]
2	Zn/TPA-TCIF(BD)	TBAB	0.5	40	10	98.8	10 (84)	[37]
3	Co@H-POP-4,4’-bipyridine	TBAB	0.3	30	48	97.1	none	[38]
4	Py-Zn@IPOP_I_	none	2	120	6	96	5 (94)	[39]
5	NHC-CAP-1(Zn^2+^)	none	2	100	3	97	10 (95)	[40]
6	ZnTPP/QA-azo-PiP_1_	none	1	80	12	99	7 (92)	[41]
7	POF-Zn^2+^-I^-^	none	1	60	8	92.2	none	[42]
8	AlPor−PIP−Br	none	0.5	40	24	98	6 (97)	[43]
9	Al-CPOP	none	0.1	120	24	95.0	5 (95)	[44]
10	Co-HIP	none	0.1	80	20	96	5 (95)	[45]
11	POP-Ni	none	0.1	80	24	97.5	6 (91.5)	This work

^a^ The yields of the cyclic carbonate from the last run are shown in the parentheses.

**Table 3 materials-16-02132-t003:** Screening on the substrate scope for CO_2_ cycloaddition with the POP-Ni catalyst ^a^.

Entry	Epoxide	Product	Yield (%) ^b^	Selectivity (%) ^b^
1	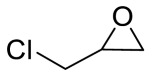	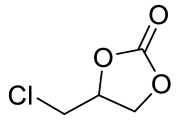	97.5	>99
2	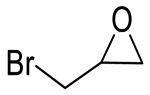	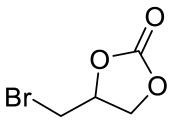	95.7	>99
3	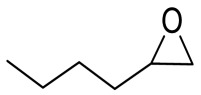	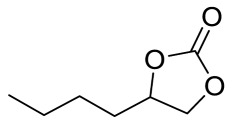	62.7	> 99
4	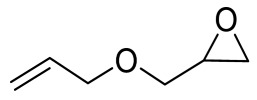	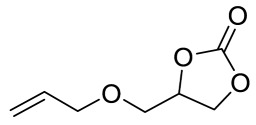	48.0	89.4
5	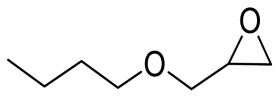	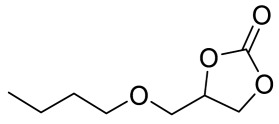	49.2	90.3
6	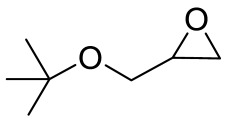	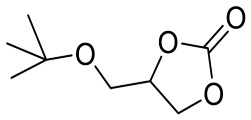	35.6	91.3
7	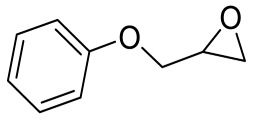	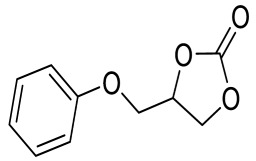	51.5	93.4
8	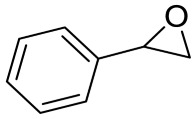	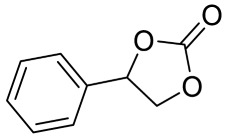	48.2	87.6

^a^ Reaction condition: 5 mmol of epoxide, 80 mg of POP-Ni, 0.5 mL of DMF, CO_2_ (balloon), 80 °C, and 24 h. ^b^ Determined by GC-MS.

## Data Availability

All data reported in this paper are contained within the manuscript.

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
