# Peer review of "A Novel POP-Ni Catalyst Derived from PBTP for Ambient Fixation of CO2 into Cyclic Carbonates"

_materials, 2023, doi:10.3390/ma16062132_

Round 1

Reviewer 1 Report

The authors aim to support a homogeneous [Ni(Me6Tren)X]X (X = I, Br, Cl) type catalyst for CO2 conversion into a PBTP in order to improve its reusability. They obtained very good results of the conversion yield, selectivity and reusability.  Evaluation of the performance of the POP-Ni catalyst found it was able to catalyze the CO2 cycloaddition with epichlorohydrin in DMF, affording 97.5% yield with 99% selectivity of chloropropylene carbonate under ambient conditions (80 °C, CO2 balloon).

The article is well structured, the obtained catalyst was deeply characterized and the results are presented correctly. My specific comments are concerning following items:

1.     In the abstract I am missing numerical data of the similar CO2 reaction with the catalyst reported in the literature. How your catalyst compares to literature?

2.     In my opinion the section 2.6. is not necessary as it is a copy of 2.5. with addition of one word. I suggest you to add a phrase to 2.5. stating that the same conditions were used for the reusability studies.

3.     Check the phrase in line 157-158 I don’t understand what it is adding to the discussion in the frame of BET results analysis. Maybe rephrase it because it is not clear taking into account the lines before.

4.     Add the results of the conversion and selectivity obtained from your previous studies on the unsupported catalyst as comparison, for example in Section 3.2.

5.     If possible, add the diameters of obtained catalyst particles based on Figure 2 a I b.

6.     Based on Table 2 I assume there were other products of the reaction as the selectivity was a bit lower than 100%. Could you state them? Add it to supplementary information.

7.     Finally, I would recommend performing more reusability studies, because based on only 4 cycles is hard to obtain clear conclusions.

Reviewer 2 Report

In this work, the authors immobilized a homogeneous catalyst and used it to catalyze the CO2 cycloaddition with epichlorohydrin in DMF. This is an interesting work, however the following issues have to be addressed prior to possible publication:

1. Abstract:

a) Make the first sentence more concise and specific to the topic of the paper.

b) Explain all abbreviations the first time used (POP, PBTP, DMF).

c) Improve language and remove informalities.

d) Change “4th” into “fourth”.

e) The last sentence has no meaning; eliminate it, and instead of it, highlight the novelty and the significance of the work.

2. Introduction:

a) Line 28: Increase of CO2, where? Be more specific.

b) Line 32: What “C1” stands for?

c) Lines 33-34: Do not use compound references. Provide reference after each type of compounds. Moreover, are fifteen references actually required here?

d) Lines 34-38: Add references.

e) Line 41: Do not use compound references.

f) Lines 44, 46, 49, 52, 61: Again, many references provided without any added value.

g) Despite the numerous references used, the literature review is poor without providing insights into CO2 conversion. Similar efforts to move from homogeneous to heterogeneous catalysis should be critically presented. The authors could summarize previous works regarding catalysts, conditions, and CO2 conversion in a Table. In this way, their results could be compared with literature.

3. Materials and Methods:

a) Line 94: Remove the reference from the heading and place it in the text appropriately.

b) Provide details about GC-MS analysis.

c) How many times was each experiment repeated. No information about the experimental error is provided.

4. Results and discussion:

a) The heading of section 3.2 needs to be changed. The authors confuse the efforts to improve the catalyst formulation in terms of activity with optimum reaction conditions. In any case, the procedure followed is not optimization.

b) Fig. S3: Were the experiments repeated? Error bars should ideally have been provided.

c) Fig. 4: Use appropriate values for y-axis. Otherwise, the observations presented in lines 269-270 cannot be verified by the reader.

d) It seems that the catalyst does not lose any activity during the first three successive runs within a reasonable experimental error. They should compare this result with literature.

e) It is not clear how the proposed mechanism in section 3.5 was derived. Was it based on GC-MS analysis results obtained? Was the technique applied able to provide all products? Without the details of the GC-MS analysis, the discussion on the mechanism is weak. Maybe, the milder “reaction pathway” could be used instead of “reaction mechanism”. The authors should provide more details and use relevant literature too. DFT could also be helpful here.

Reviewer 3 Report

This paper by Wang and Xie reports a synthetic methodology for the preparation of a heterogeneous bifunctional POP-Ni catalyst, through the copolymerization of 4,4'-(bromomethyl) biphenyl (BBMBP) and 1,3,5-tris (bromomethyl) (TBB) via Friedel-Crafts alkylation, followed by a post-synthetic modification procedure in two-steps, which involved the grafting Me6Tren on the PBTP and the subsequent coordination of Ni(II) salt. The new material was characterized by appropriate techniques (SEM, XPS, N2-sorption, TGA, ICP-MS) and applied in CO2 cycloaddition reaction to epoxides, leading to active and reusable heterogeneous catalysts in the absence of any co-catalyst.

The topic is relevant and the paper is well written and well-organized. In my opinion, this paper can be accepted in Materials after minor revisions. Dome comments are presented bellow:

11) In order to clearly understand the relevance of this study, the authors should highlight comparative results with other reported catalysts immobilized in POP supports in the same reaction.

22)The Conclusions should contain more detailed information (with numbers) and should be improved in order to highlight the significance of the work to the current state of the art.

33) Finally, studies of CO2 adsorption/desorption isotherms could provide relevant information regarding CO2-support interaction and establish a better correlation between the porous structure and catalytic activity.

Round 2

Reviewer 2 Report

The authors addressed almost all issues raised. Just one comment for them: The review process is not a personal dialogue between the authors and the reviewers but an effort to improve the quality of papers. It is a common practice to repeat experiments and report error bars as one standard deviation (the simplest way), particularly when the final target value is very similar from one experiment to another. Deviations are inherent in most processes and referring to previous works actually means nothing regarding reporting data.